# Probiotics Improve Eating Disorders in Mandarin Fish (*Siniperca chuatsi*) Induced by a Pellet Feed Diet via Stimulating Immunity and Regulating Gut Microbiota

**DOI:** 10.3390/microorganisms9061288

**Published:** 2021-06-12

**Authors:** Xiaoli Chen, Huadong Yi, Shuang Liu, Yong Zhang, Yuqin Su, Xuange Liu, Sheng Bi, Han Lai, Zeyu Zeng, Guifeng Li

**Affiliations:** 1State Key Laboratory of Biocontrol, Guangdong Provincial Key Laboratory for Aquatic Economic Animals, School of Life Sciences, Sun Yat-Sen University, Guangzhou 510275, China; chenxli27@mail2.sysu.edu.cn (X.C.); yihd3@mail2.sysu.edu.cn (H.Y.); liush276@mail2.sysu.edu.cn (S.L.); lsszy@mail.sysu.edu.cn (Y.Z.); suyq9@mail2.sysu.edu.cn (Y.S.); liuxg27@mail2.sysu.edu.cn (X.L.); bish@mail2.sysu.edu.cn (S.B.); laih5@mail2.sysu.edu.cn (H.L.); zengzy5@mail3.sysu.edu.cn (Z.Z.); 2Guangdong Provincial Engineering Technology Research Center for Healthy Breeding of Important Economic Fish, Guangzhou 510006, China

**Keywords:** eating disorders, feeding behavior, gut microbiota, *Siniperca chuatsi*, innate immunity, appetite, *Lactobacillus plantarum*, *Lactobacillus rhamnosus*, *Clostridium butyricum*

## Abstract

Eating disorders are directly or indirectly influenced by gut microbiota and innate immunity. Probiotics have been shown to regulate gut microbiota and stimulate immunity in a variety of species. In this study, three kinds of probiotics, namely, *Lactobacillus plantarum*, *Lactobacillus rhamnosus* and *Clostridium butyricum*, were selected for the experiment. The results showed that the addition of three probiotics at a concentration of 10^8^ colony forming unit/mL to the culture water significantly increased the ratio of the pellet feed recipients and survival rate of mandarin fish (*Siniperca chuatsi*) under pellet-feed feeding. In addition, the three kinds of probiotics reversed the decrease in serum lysozyme and immunoglobulin M content, the decrease in the activity of antioxidant enzymes glutathione and catalase and the decrease in the expression of the appetite-stimulating regulator agouti gene-related protein of mandarin fish caused by pellet-feed feeding. In terms of intestinal health, the three probiotics reduced the abundance of pathogenic bacteria *Aeromonas* in the gut microbiota and increased the height of intestinal villi and the thickness of foregut basement membrane of mandarin fish under pellet-feed feeding. In general, the addition of the three probiotics can significantly improve eating disorders of mandarin fish caused by pellet feeding.

## 1. Introduction

Eating disorders mainly refer to a group of syndromes characterized by abnormal feeding behaviors, accompanied by significant weight changes or physiological dysfunctions [1,2,3]. The main clinical types include anorexia nervosa, bulimia nervosa, binge eating disorder and avoidance/restrictive food intake disorder. Moreover, eating disorders occur throughout the age groups and have an essential impact on physical and mental health [4]. They increase the likelihood of anxiety, obesity, suicidal intentions, depression, drug abuse and health problems [5]. Eating disorders are associated with the establishment of food preferences and aversions and are influenced by the sensorial characteristics of food [6]. A better understanding of food preferences and aversions can improve the prevention and treatment of eating disorders [7].

Food preference is an innate behavioral trait which is affected by both genes and the environment [8,9]. The hypothalamus contains orexigenic neurons that express neuropeptide Y (NPY) and agouti-related peptide (AgRP), which participate in food intake control and are regulated by the peripheral hormone leptin and ghrelin [10,11]. NPY is a peptide composed of 36 amino acids. As an appetite-stimulating factor, it plays a crucial role in regulating energy homeostasis and food intake [12]. AgRP increases food intake by antagonizing the effect of the anorexigenic POMC product, α-melanocyte stimulating hormone (α-MSH) [13,14]. There seems to be a species-specific variability in the functions of leptin and ghrelin with regards to the regulation of feeding and metabolism in fish [15]. Ghrelin acts as an appetite stimulant in a variety of fish species, but there is also conflicting evidence, such as in *Salmoniformes* [16].

As an essential modulator of host physiology and behavior, intestinal bacteria have been shown to influence feeding behavior and food choice [17,18,19,20,21,22,23,24,25]. Gut microbiota can influence host eating behavior by directly affecting nutrient sensing, appetite and satiety-regulating systems through the production of neuroactive substances and short-chain fatty acids or indirectly manipulating intestinal barrier function, interacting with bile acid metabolism, modulating the immune system and influencing host antigen production [26]. Gut microbiota play a vital role in regulating host eating disorders’ behavioral comorbidities, such as obesity, anorexia nervosa and severe acute malnutrition. A growing body of evidence links the gut microbiota with nutrition, immune, anti-oxidative stress and appetite. Influencing one of these factors will most likely lead to changes in the others, thereby making the gut microbiota easily accessible and manipulable for targeting host food preferences [26].

Administration of probiotics is an effective strategy to maintain the balance of the gut microbiota [27]. Probiotics are defined as microbial cells or compounds that have a beneficial effect on the health of the host. In aquaculture, probiotics can prevent the spread of diseases, increase food conversion efficiency and stimulate growth by improving the composition of the gastrointestinal microbiota, strengthening the immune system and increasing the resistance to farmed stressors [28,29,30]. In addition, probiotics have become an alternative to antibiotics and other drug treatments in the aquaculture industry and are considered a new tool for disease control [28,31,32]. Microorganisms commonly used as probiotics in aquaculture include bacteria, yeast and algae [33].

Among several probiotic bacterial species, numerous reports have been published on the beneficial role of *Lactobacillus plantarum*, *Lactobacillus rhamnosus* and *Clostridium butyricum* as probiotics in aquaculture [34,35,36,37,38,39,40]. *L. plantarum* is a rod-shaped, gram-positive, non-spore-forming facultative anaerobic bacteria that belong to the *Lactobacillaceae* family. It has been reported to reduce the adhesion and growth of harmful bacteria via producing antimicrobial compounds [41,42,43], improve the growth and feed efficiency of carp (*Catla catla*) [44,45], grouper (*Epinephelus coioides*) [46], tilapia (*Oreochromis niloticus*) [47], shrimp (*Penaeus indicus*) [48] and pacific white shrimp (*Litopenaeus vannamei*) [49] and enhance the immunity and survival rate of pacific white shrimp (*Litopenaeus vannamei*) [50,51] and tilapia [52]. Previous studies have shown that *L. rhamnose* can affect the appetite and energy metabolism of the host by regulating the expression of γ-aminobutyric acid and its receptors in the central nervous system [53,54,55,56,57]. *C. butyricum* is a spore-forming bacterium belonging to Gram-positive anaerobe that can produce butyric acid and exists in the intestine of healthy animals and human [58,59,60]. Compared with other probiotics, *C. butyricum* has a more vital tolerance ability to higher temperature environments, lower pH, bile salt and several antibiotics. Therefore, *C. butyricum* has always been regarded as a good and safe food additive [58]. *C. butyricum* has a positive effect on immune function and is connected with increased population of *Bifidobacterium* and *Lactobacillus* and decreased concentration of pathogenic bacteria in the intestinal tract of humans, mice, piglets and broiler chickens [61,62]. *C. butyricum* can inhibit intestinal inflammation and regulate gut microbiota through the immune pathway [63,64,65].

Mandarin fish (*Siniperca chuatsi*) is a precious freshwater farmed fish with unique live bait feeding habits, and it does not easily accept dead bait or pellet feed [10,66]. The preference for a live bait diet increases the cost of mandarin fish farming and the risk of infectious diseases, limiting the development of mandarin fish farming. For this problem, previous studies mainly focused on optimizing the domestication process and breeding conditions (such as temperature), strengthening the training of learning and memory and using attractants, which promoted the development of pellet feed for mandarin fish [67,68,69,70]. However, there are still problems such as high mortality and slow growth of mandarin fish fed with pellet feed. Recently, relationships among gut microbiota, host immunity and feeding preference behavior have attracted research attention [71]. Probiotics intervention is an effective way to regulate the gut microbiota [27]. In this study, three probiotics that have been shown to be safe for aquatic animals, *L. plantarum*, *L. rhamnosus* and *C. butyricum*, were selected to investigate whether probiotics can improve the eating disorders of mandarin fish caused by pellet feed diet by modulating the gut microbiota, immune parameters, appetite and intestinal morphology, which may contribute to the theoretical foundation of probiotics intervention in the treatment of dietary disorders.

## 2. Materials and Methods

### 2.1. Bacteria Strains

The three probiotic strains, *L. plantarum* (ATCC 8014), *L. rhamnosus* (ATCC 7469) and *C. butyricum* (ATCC 19398), were purchased from Guangdong Microbial Culture Collection Center (GDMCC). The bacteria were cultured as described previously [36,72,73]. Briefly, the two activated bacterial suspensions of *L. plantarum* and *L. rhamnosus* were separately incubated into MRS liquid broth (Merck, Darmstadt, Germany). The activated bacterial suspension of *C. butyricum* was incubated into the reinforced clostridial medium (RCM) and then placed in an anaerobic workstation at 37 °C for 12 h. The bacterial titers were measured by making tenfold dilution series in triplicate on agar plates. Optical densities (OD) were measured using a spectrophotometer (Spectroscan UV 2600, Thermo Scientific, Waltham, MA, USA) at 600 nm. The strains were harvested via centrifugation at 4000× *g* for 10 min, washed twice with normal saline (0.9% NaCl) and resuspended at 2 × 10^10^ colony forming unit (CFU)/mL in sterile normal saline. Culture bacterial cells were afterward kept at 4 °C until usage.

### 2.2. Animal Treatments

All experimental procedures were approved by the Institutional Animal Care and Use Committee of Sun Yat-sen University and performed according to the guidelines for experimental animals established by this committee. One thousand and five hundred healthy mandarin fish were obtained from a fish farm in Foshan, Guangdong, China. All experimental fish were acclimatized for two weeks in 3200 L rectangular aquaria to laboratory conditions before pellet-feed feeding.

After the adaptive feeding, a total of 1350 healthy mandarin fish weighing 2.5 ± 0.1 g (mean ± standard error of mean (SEM)) were randomly allocated into one of five groups (270 fish per group): live bait fish feeding group (LBFD), pellet-feed feeding group with probiotics free (PFD), pellet-feed feeding group with *L. plantarum* plus (PFDLP), pellet-feed feeding group with *L. rhamnosus* plus (PFDLR) and pellet-feed feeding group with *C. butyricum* plus (PFDCB). Each group of experimental fish was randomly assigned to three 800 L replicated water tanks (90 fish per tank). Mandarin fish in the PFDLP, PFDLR and PFDCB groups were treated with *L. plantarum*, *L. plantarum* and *C. butyricum* at a final concentration of 10^8^ CFU/mL for one week, while the remaining two groups, LBFD and PFD, were not treated. In this time, all fish received a live bait fish diet twice a day (at 06:00 and 18.00 h) at 5% of initial body weight. Mud carp (*Cirrhinus molitorella*) was used as the live bait fish in this study.

During the period of pellet-feed feeding, the PFD, PFDCB, PFDLR and PFDLP groups of experimental fish were overfed from dead fish (1 week) to commercial feed (4 weeks) following the domestication process established by Liang et al. [67], while the LBFD group of experimental fish maintained a live bait diet. Each group of experimental fish was fed twice a day (at 06:00 and 18.00 h) at 5% of initial body weight to approximate satiation. The main nutritional composition of the commercial feed purchased from Foshan Nanhai Jieda Feed Co., LTD. (Lishui, China), is 48% crude protein, 5% crude fat, 3% crude fiber, 19% crude ash, 10% water, 4% calcium, 2% total phosphorus, 3%NaCl and 2.7% lysine. The soft pellet feed with a diameter of 50 mm was made with a feed machine and stored at −20 °C until use. Part of the water tank was replaced daily to remove waste and feces. When partially replacing the aquaculture water, an appropriate amount of *L. plantarum*, *L. plantarum* and *C. butyricum* was added to the PFDLP, PFDLR and PFDCB groups to maintain the concentration at 1 × 10^8^ CFU/mL. The water quality of each tank was kept within the best physical parameter range, temperature (24.13 ± 0.52 °C), pH (7.41 ± 0.15), ammonia-nitrogen (0.27 ± 0.05 mg/L) and dissolved oxygen (7.52 ± 0.15 mg/L), during the experiment.

### 2.3. Proportion of Pellet Feed Recipients and Survival Analysis

The number of pellet feed recipients in groups PFD, PFDLP, PFDLR and PFDCB were counted on days 7, 14 and 28 after pellet-feed feeding, and the proportion of pellet feed recipients (POPFR) was calculated according to the following formula: POPFR (%) = [Number of pellet feed recipients/Number of initial mandarin fish] × 100. During the feeding trial, the number of deaths in each group was recorded every day, and Kaplan Meyer’s (KM) survival analysis was used to evaluate the survival differences between groups.

### 2.4. Sample Collection

On days 7, 14 and 28 of pellet-feed feeding, twelve mandarin fish were randomly collected from each tank and then anesthetized with tricaine methanesulfonate (MS-222) for subsequent sampling. Blood samples collected from the tail vascular vein of each fish were placed in centrifuge tubes and centrifuged at 4 °C and 4000 rpm for 15 min to separate the serum. The separated serum was stored at −80 °C for further determination of immune parameters. Brain and gut samples were collected and placed in RNA Later^®^ (Qiagen, Hilden, Germany) at 4 °C overnight and then stored at −80 °C for gene expression analysis. A separate liver, intestine and gills were homogenized with cold phosphate buffer saline (PH 7.5). The homogenate was then centrifuged at 4 °C and 8000 rpm for 10 min, and the supernatant was taken and stored at −20 °C for analysis of antioxidants and oxidative stress parameters. Intestinal samples containing the inclusion were collected and placed in sterile Eppendorf tubes, immediately frozen in liquid nitrogen, and then stored at −80 °C for microbiome analysis. Intestinal tissue was collected and fixed in Bouin’s solution for 24 h before histological analysis was performed.

### 2.5. Serum Parameter Analysis

#### 2.5.1. Serum Lysozyme Content

According to the instruction manual, lysozyme content in serum was strictly analyzed (Nanjing Jiancheng Bioengineering Institute, Nanjing, China).

#### 2.5.2. Measurement of IgM and CRP

Reagent kits for immunoglobulin M (IgM) and C-reactive protein (CRP) were obtained from Shanghai Enzyme-linked Biotechnology Co., Ltd., Shanghai, China. Each parameter was strictly analyzed in accordance using a double-antibody sandwich ELISA with the manufacturer’s instructions.

### 2.6. Antioxidant and Oxidative Stress Parameters

The superoxide dismutase (SOD) activity, CAT activity, glutathione (GSH) content and malondialdehyde (MDA) content were determined according to the instructions provided in the commercial kits (Nanjing Jiancheng Bioengineering Institute, Nanjing, China). SOD, GSH, CAT and MDA measurements were based on the WST-1 method [74], xanthine oxidase method [75], ammonium molybdate colorimetric method [76] and thiobarbituric acid method [77], respectively.

### 2.7. Gene Expression Analysis

#### 2.7.1. Extraction of total RNA and Reverse Transcription

According to the manufacturer’s instructions, total RNAs were extracted from each tissue sample (50–100 mg) using RNAiso Plus reagent (Takara, Shiga, Japan). RNA concentrations and purity were determined using a Nanodrop 2000 c spectrophotometer (Thermo Fisher, Waltham, MA, USA). RNA was used as a templet for cDNA synthesis using PrimeScript^TM^ reverse transcription (RT) reagent kit (TaKaRa, Shiga, Japan) following the manufacturer’s guidelines and stored at −80 °C until analysis.

#### 2.7.2. Real-Time Quantitative PCR (RT-qPCR)

Total RNA was isolated from different tissues by using RNAiso Plus reagent (Takara, Shiga, Japan) according to the manufacturer’s instructions. First-strand complementary DNAs (cDNAs) were synthesized using PrimeScript^TM^ RT reagent kit (Takara, Shiga, Japan) following the manufacturer’s guidelines. The expression levels of *ghrelin*, *leptin*, *npy*, *agrp* and *β-actin* were detected using the corresponding forward and reverse primers, which were designed using Primer Express software (Applied Biosystems, Waltham, MA, USA) (Table 1). *β-actin* served as a housekeeping gene in order to normalize the expression levels. Quantitative PCR (qPCR) was performed on a total reaction volume of 10 μL, containing 0.2 μM primers, 1μL of cDNA, 5 μL of 2 × SYBR premix ExTaq™ (Takara, Shiga, Japan) and 3.6 μL of ultrapure water using the following setting: 40 cycles of amplification (5 s at 95 °C, 40 s at 60 °C and 1 s at 70 °C). All RT-qPCR reactions were performed in triplicate on a LightCycler 480 instrument (Roche Diagnostics, Rotkreuz, Switzerland). Data were analyzed using the 2^-ΔΔCt^ method [78].

### 2.8. Gut Microbiota Analysis

Total bacterial DNA of the intestine samples with retained contents was extracted using an E.Z.N.A. ^®^Stool DNA Kit (Omega, Norcross, GA, USA). After measurement of the concentration and quality of the extracted DNA using a Nanodrop 2000c spectrophotometer (Thermo Fisher, Waltham, MA, USA), the V4-V5 region of the bacterial 16S DNA gene was amplified via the PCR method using the primers of 515F (5′-GTGCCAGCMGCCGCGGTAA-3′) and 806R (5′-CCGTCAATTCCTTTG AGTTT-3′). The high throughput sequencing for the qualified amplicon was performed on the Illumina NovaSeq6000 platform at Novogene Biotech Co., Ltd. (Beijing, China). Paired-end reads were assigned to samples based on a unique barcode and truncated by cutting off the barcode and primer sequence. The raw tags were then produced via FLASH (V1.2.7) [79]. Sequences were analyzed with the UCHIME algorithm [80] and QIIME [81]. The effective tags were filtered and clustered into operational taxonomic units (OTUs) under a 97% nucleotide similarity level. The taxonomic annotation of OTUs was performed using Uparse software [82]. The alpha diversity, including the observed species, Chao 1, abundance-based coverage estimator (ACE), Simpson, Shannon and PD whole tree, was calculated using QIIME (Version 1.9.1) to analyze the abundance and diversity. A Venn diagram was constructed to describe the core components of the genera. Beta diversity was evaluated using principal coordinates analysis (PCoA). Linear discriminant analysis effect size (LEfSe) was used to identify significant differences in the relative abundance of bacterial taxa [83]. Predicted functional pathways were annotated using the Kyoto encyclopedia of genes and genomes (KEGG) at level 1. Tax4Fun was used to predict the functional profile of the intestinal microbiota [84]. All figures were drawn using R software (Version 2.15.3).

### 2.9. Intestinal Histological Assessment

The foregut, midgut and hindgut tissues were fixed in Bouin’s solution for 24 h and then dehydrated, embedded in paraffin and sectioned into 4-μm transverse cuts following the axis of the gut lumen. Hematoxylin and eosin (H.E.) were applied for the staining, and histological examination of the samples was carried out using an optic microscope (Nikon, Tokyo, Japan) with a digital camera (Nikon, Tokyo, Japan). The intestinal villi height and basement membrane thickness of each segment was measured with Image-Pro software.

### 2.10. Statistical Analysis

All the experimental data were tested for normality and homogeneity of variances using the Shapiro-Wilk’s test and Levene’s test, respectively, and presented as the mean ± SEM. Significant differences were determined using the one-way analysis of variance (ANOVA) test, followed by Fisher’s least significant difference post hoc test and Duncan’s multiple range tests, after confirming data normality and homogeneity of variances. Statistical analysis was performed using SPSS software 19.0 (SPSS Inc., New York, NY, USA) and the Windows-based Graph pad prism statistical software (San Diego, CA, USA). A *p* value less than 0.05 was accepted as statistically significant.

## 3. Results

### 3.1. Proportion of Pellet Feed Recipients

The POPFR of mandarin fish in different feeding groups (PFD, PFDCB, PFDLR and PFDLP) was tested on the 7th, 14th and 28th day of feeding. As shown in Figure 1, on the 28th day of feeding, the POPFR of mandarin fish in the PFDLP, PFDLR and PFDCB groups was higher than that in the PFD group, and the PFDLP and PFDCB groups reached a significant level of difference (*p* < 0.05). The highest POPFR of mandarin fish was recorded in PFDLP (81%) compared to PFD (68%) on the 28th day of feeding.

### 3.2. Survival Analysis

Mandarin fish fed with pellet feed without probiotics supplemented had a lower survival rate than those fed with live bait at the end of the experiment (Figure 2). Application of *L. plantarum*, *L. rhamnosus* and *C. butyricum* significantly reduced the decrease of the survival rate of mandarin fish caused by the pellet feed diet at the end of the experiment (Figure 2).

### 3.3. Serum Parameter Analysis

#### 3.3.1. Serum Lysozyme Content

Mandarin fish in the PFD group had lower serum lysozyme content than that in the LBFD group at days 7, 14 and 28 of feeding (Figure 3). The effects of *L. plantarum*, *L. rhamnosus* and *C. butyricum* on serum lysozyme content are shown in Figure 3. Application of *L. plantarum*, *L. rhamnosus* and *C. butyricum* reduced the decrease of the serum lysozyme content of mandarin fish caused by the pellet feed diet (Figure 3). Compared with the PFD group, the content of serum lysozyme increased significantly on the 7th and 28th day in the PFDLP group and on the 14th day in the PFDLR group (*p* < 0.05) (Figure 3). The highest serum lysozyme content of mandarin fish was noticed in PFDLP after being fed for 28 days (Figure 3).

#### 3.3.2. Measurement of IgM and CRP

The effects of *L. plantarum*, *L. rhamnosus* and *C. butyricum* on serum IgM and CRP content are shown in Figure 4. Although the application of *L. plantarum*, *L. rhamnosus* and *C. butyricum* reduced the decrease of the serum IgM level of mandarin fish at days 14 and 28 of feeding (Figure 4A), serum CRP content was not significantly affected by pellet feed and probiotics application (Figure 4B). Compared with the PFD group, the serum IgM content of the PFDLP group supplemented with *L. plantarum* was significantly increased at days 14 and 28 of feeding (*p* < 0.05) (Figure 4A).

### 3.4. Antioxidant and Oxidative Stress Parameters

GSH content and CAT activity in liver, gut and gill of mandarin fish in the PFD group were decreased compared with that in the LBFD group (Figure 5B,C), while the MDA level in liver, gut and gill was increased (Figure 5D). Compared with the LBFD group, the decrease of GSH content in gut, the decrease of CAT activity in gill and the increase of MDA content in gill in the PFD group reached significant difference levels (*p* < 0.05) (Figure 5B–D). The application of *L. plantarum*, *L. rhamnosus* and *C. butyricum* motivated an elevation of GSH content and CAT activity (Figure 5B,C) and a reduced MDA content in the liver, gut and gill of mandarin fish in the PFDLP, PFDLR and PFDCB groups when compared to the PFD group (Figure 5D). The content of GSH in liver and gill of mandarin fish in the PFDLP group treated with *L. plantarum* was significantly higher than that in the PFD group (*p* < 0.05) (Figure 5B). Compared with the mandarin fish in the PFD group, application of *L. plantarum*, *L. rhamnosus* and *C. butyricum* significantly increased CAT activity in liver and MDA content in gill (*p* < 0.05) (Figure 5C,D).

### 3.5. Expression of Appetite-Related Genes

For appetite control genes expression in the brain and gut of mandarin fish in different feeding groups (LBFD, PFD, PFDCB, PFDLR and PFDLP) after being fed for 28 days, we found a significantly increased mRNA level of *leptin* in the gut of mandarin fish and a significantly decreased mRNA level of *npy* and *agrp* in the brain of mandarin fish in the PFD group compared to the LBFD group (*p* < 0.05) (Figure 6A,B). After applying *L. plantarum*, *L. rhamnosus* and *C. butyricum*, the *leptin* expression levels in the mandarin fish gut were significantly down-regulated in the PFDLP and PFDCB groups compared with that in the PFD group (*p* < 0.05) (Figure 6A,B). The *agrp* expression levels in the mandarin fish brain were significantly up-regulated in the PFDLP, PFDLR and PFDCB groups compared with that in the PFD group (*p* < 0.05) (Figure 6A,B).

### 3.6. Gut Microbiota Analysis

#### 3.6.1. Richness and Diversity

The alpha diversity index, including observed species, Shannon, Simpson, Chao 1, ACE and PD whole tree, was calculated to assess the diversity and richness of intestinal microbiota of mandarin fish in different groups. No significant difference was observed in the Shannon and Simpson indices between groups (*p* < 0.05) (Table 2). The observed species, Chao1, ACE and PD whole tree indices of the PFDLP group were higher than that of other groups, and there was significant difference compared with the LBFD and PFDCB groups (*p* < 0.05) (Table 2). A Venn diagram was constructed to identify the core and different OTUs existing in mandarin fish under different feeding strategies. In this regard, 168 OTUs were shared among all mandarin fish gut samples. In contrast, 430 OTUs, 534 OTUs, 661 OTUs, 269 OTUs and 176 OTUs were unique to LBFD, PFD, PFDLP, PFDLR and PFDCB groups, respectively (Figure 7). Simultaneously, the intestinal microbiota community structure was further investigated using PCoA based on the binary jaccard distance (Figure 8). PCoA analysis showed 16.8% and 12.16% explained variance of principal component analysis PCoA1 and PCoA2, respectively. PCoA cluster analysis indicated that three clusters were formed and separated between the bait fish diet group (LBFD), pellet feed group (PFD) and probiotic-treated pellet feed group (PFDLP, PFDLR and PFDCB) after being fed for 28 days (Figure 8). This suggested that different feeding strategies of mandarin fish led to different intestinal community structures (Figure 8).

#### 3.6.2. Community Composition and Biomarker Analysis

The gut microbiota of mandarin fish in different feeding groups (LBFD, PFD, PFDCB, PFDLR and PFDLP) showed their unique microbial population structure. At the phylum and genus level, the top 10 abundant microbiota composition in the intestine of mandarin fish in different feeding groups (LBFD, PFD, PFDCB, PFDLR and PFDLP) after being fed for 28 days is represented in Figure 9. The gut microbiota of mandarin fish in the LBFD group was dominated by *Fusobacteriota* and *Proteobacteria* at the phylum level, and *Proteobacteria* was the dominant phylum in the gut microbiota of the PFD, PFDLP, PFDLR and PFDCB groups (Figure 9A). The abundance of *Aeromonas* in the PFDLP, PFDLR and PFDCB groups was significantly lower than in the PFD group (Figure 9B). LEfSe analysis revealed 19, 24, 17, 7 and 1 biomarkers with significantly higher relative abundance in the LBFD, PFD, PFDLP, PFDLR and PFDCB groups, respectively (Figure 10A). *Aeromonas* was a biomarker for PFD compared with other groups (Figure 10B).

#### 3.6.3. Functional Prediction

Functional prediction on the KEGG database was annotated based on 16S sequencing data. As shown in Figure 11, the abundance of functional categories based on KEGG (level 1) between different feeding groups (LBFD, PFD, PFDCB, PFDLR and PFDLP) after being fed for 28 days were analyzed. The abundance of human pathogens pneumonia and human pathogens nosocomial significantly increased in the PFD group compared with other groups (*p* < 0.05) (Figure 11).

### 3.7. Intestinal Histological Assessment

Histological changes of the intestinal tract were observed in different feeding groups (LBFD, PFD, PFDCB, PFDLR and PFDLP) after being fed for 28 days (Figure 12). By comparing LBFD, PFD, PFDCB, PFDLR and PFDLP, the result showed that *C. butyricum*, *L. rhamnosus* and *L. plantarum* could significantly increase the villi height of the foregut, midgut and hindgut of mandarin fish fed with pellet feed (*p* < 0.05; Figure 13A) and significantly reverse the decrease in the thickness of foregut basement membrane caused by pellet-feed feeding (*p* < 0.05; Figure 13B).

## 4. Discussion

Mandarin fish have a food preference for live bait and show certain eating disorders with dead bait fish or pellet feed. The increased mortality rate of mandarin fish under pellet feeding conditions seriously affects its economic benefits [10,66]. The eating disorder is characterized by abnormal feeding behaviors associated with the establishment of food preference [1,2,3,6]. Gut microbiota can regulate host food preferences through interactions with nutritional, immune, antioxidant stress and appetite levels [26,85]. Previous studies have shown that probiotics can influence the feeding behavior of the host by regulating the microbiota [26,27,86]. However, few studies have been done on the regulatory effect of probiotics on eating disorders, especially on the pellet feed intake of mandarin fish [87]. Therefore, the present study was conducted to assess the effects of *L. plantarum*, *L. rhamnosus* and *C. butyricum* on the POPFR, survival, appetite, gut microbiota, innate immunity, antioxidant capacity and intestinal histology in mandarin fish and to explore the role of probiotics in regulating feeding behavior in vivo.

Acceptance of pelleted feed and survival rate are direct indicators of the improvement of eating disorders during the feeding process of mandarin fish with pellet feed. In this study, we observed that supplementation with either of the three probiotics effectively increased the POPFR in mandarin fish compared to those fed the same diet but without probiotics supplementation. Moreover, pellet feed diet can lead to the reduction of survival rate of mandarin fish, which is consistent with the previous report that the dietary conversion of *Sparus aurata* larvae and *Solea senegalensis* larvae from live bait to alginate microdiets resulted in a significant decrease in survival rate, which may be related to the changes of physiological stress and nutritional status of the larvae fish [88,89,90]. Results showed that *L. plantarum*, *L. rhamnosus* and *C. butyricum* can significantly reverse the increase in mortality of mandarin fish caused by feeding pellets at the end of the 28-day experiment. This finding is consistent with a previous study in which the administration of *L. plantarum* to the rainbow trout at a dose of 10^6^ CFU/g for 36 consecutive days significantly improved the survival rate of rainbow trout when attacked by *Lactococcus garvieae* [91]. Similarly, Hooshyar reported that *L. rhamnosus* ATCC 7469 significantly increased the survival rate of rainbow trout (*Oncorhynchus mykiss*) when attacked by *Yersinia ruckeri* [36]. Duan reported that supplementation of *C. butyricum* (1 × 10^9^ CFU/g) for 56 days improved the survival of black tiger shrimp (*Penaeus monodon*) after exposure to nitrite stress for 24 and 48 h [92]. Proper nutrition can affect intestinal health through several pathways, including intestinal morphology, microbial diversity, intestinal barriers and oxidative status [93]. The improvement of survival of cultured animals after applying *L. rhamnosus*, *L. plantarum* and *C. butyricum* may result from their positive regulation of nutritional status, intestinal morphology, gut microbiota, oxidative status and immune system [38,94,95]. Therefore, the administration of probiotics may be a potential method to improve the eating disorders of mandarin fish caused by pellet feed and increase the POPFR of mandarin fish without side effects because probiotics such as *L. plantarum*, *L. rhamnosus* and *C. butyricum* are generally regarded as safe for aquatic animals.

Appetite is one reason influencing the eating preference of mandarin fish [96]. Feeding behavior is ultimately regulated by central feeding centers of the brain, which receive and process information from endocrine signals from both the brain and periphery. These signals, such as hormones that inhibit (e.g., leptin) or increase (e.g., Agrp) ingestion, provide information about nutritional status and ingestion [97,98,99]. Npy is considered the most potent orexigenic molecule in fish, mediated by gut microbiota changes [100,101]. Agrp is one of the most potent appetite stimulants within the hypothalamus and mediates the peripheral body weight regulators such as ghrelin and leptin [100,102]. In the present study, we observed that *L. rhamnosus*, *L. plantarum* and *C. butyricum* could reverse the decrease of *agrp* expression in the brain tissue of mandarin fish caused by pellet-feed feeding. At the peripheral level, ghrelin is a potent appetite stimulant and is highly expressed in the fish gut [103,104]. Furthermore, the gastrointestinal hormone ghrelin is a vital molecule that regulates intestinal motility and secretion [105,106]. Leptin plays an anorexic role by down-regulating orexigenic signals such as Npy [107]. This study showed that the treatment of *L. rhamnosus*, *L. plantarum* and *C. butyricum* can reverse the high expression of the peripheral hormone leptin in the intestinal tissue of mandarin fish caused by feed-pellet feeding. These results agree with the previous findings on the regulation of appetite of *L. rhamnosus* on larval Nile tilapia [38]. All this indicates that probiotics treatment can promote the appetite of pellet feeding mandarin fish through reducing the expression of the peripheral appetite-suppressing hormone leptin and increasing the expression of the central appetite-promoting factor Npy/Agrp. Previous studies have shown that the gut microbiota can affect host appetite and eating behavior by directly affecting nutrient sensing and the satiety regulation system [26]. In this study, the appetite-promoting effect of *L. rhamnosus*, *L. plantarum* and *C. butyricum* may be mediated by their regulation on the gut microbiota of mandarin fish.

The immune system can influence eating behavior through interactions with gut bacteria and appetite [108,109]. As lower vertebrates, fish mainly rely on the innate immune system to resist pathogens [110]. Lysozyme is responsible for bacterial lysis and activation of phagocytes and complement systems [111]. IgM mainly exists in the serum, which is the most essential component of teleost humoral immunity, and it can recognize, bind and precipitate antigens and activate the complement system [112]. To assess if *L. rhamnosus*, *L. plantarum*, *C. butyricum* affects the immune system of feed-fed mandarin fish, we measured the levels of lysozyme and IgM in the serum. We found that at the end of 28 days of cultivation, the three probiotics can increase the reduction of mandarin fish serum lysozyme and IgM content caused by pellet feed domestication, and *L. plantarum* is the most significant. All this is similar to the finding in a previous publication suggesting that the feed supplement of *L. plantarum* CCFM8661 restored the decrease in serum lysozyme of Nile tilapia caused by waterborne Pb exposure [113]. In Wang’s study, administration of *C. butyricum* significantly increased the serum IgM levels in piglets on day 28 [114]. Liao et al. have confirmed that a diet supplemented with *C. butyricum* increased the IgM concentration compared with that of chicks in the control group at 21 and 42 days old [115]. This study proved that the addition of *L. plantarum*, *L. rhamnosus* and *C. butyricum* reversed the decrease in serum lysozyme and IgM content caused by pellet-feed feeding, which may further ameliorate eating disorders by regulating the appetite and gut microbiota of mandarin fish.

Anti-oxidative enzymes are the major components of anti-oxidative defense systems in living organisms [116]. The host gut microbiota directly or indirectly influences the central nervous system by affecting local OS levels and the permeability of the gut and then influences the behavioral characteristics of the host. SOD, CAT and GSH are considered the three main antioxidant enzymes in the primary antioxidant defense system, eliminating ROS in the body during oxidative damage [117]. MDA is an essential product of membrane lipid peroxidation and a well-known aging indicator reflecting the degree of oxidative stress in cells [118]. In this study, compared with the LBFD group, the decrease of GSH content in gut, the decrease of CAT activity in gill and the increase of MDA content in gill in the PFD group reached significant difference levels (*p* < 0.05), indicating that the pellet diet induced oxidative stress in mandarin fish, which is in accord with the results found in *Solea senegalensis* larvae and hybrid mandarin fish [88,119]. The low nutritional status and stress caused by the pellet diet may decrease antioxidant capacity in mandarin fish [88,120,121,122,123,124]. Furthermore, compared with the mandarin fish in the PFD group, application of *L. plantarum*, *L. rhamnosus* and *C. butyricum* significantly increased CAT activity in liver and MDA content in gill (*p* < 0.05). Increased CAT activity and GSH content accompanied by decreased MDA levels was observed after the application of three probiotics compared with the PFD group, which indicates that *L. rhamnosus*, *L. plantarum* and *C. butyricum* could enhance the antioxidant capacity of the host, which is consistent with the findings in rainbow trout, the black tiger shrimp (*Penaeus monodon*) and Nile tilapia [36,92,113,125]. The three kinds of probiotics showed an excellent free radical scavenging ability in the oxidative damage of the liver, intestine, and gill tissues, which may be attributed to its ability in gut microbiota and immune system regulation.

It has been reported that gut microbiota plays a causal role in regulating the feeding behavior of the host and can directly or indirectly affect the appetite and food intake of the host [26,126]. The composition of the intestinal microbiome is influenced by both host genotype and environment. Previous studies have shown that the gut microbiota of aquatic species is influenced by several abiotic factors [127,128]. Diet is considered one way to change the gut microbiota and the exogenous factors affecting the gut microbiota [129,130,131,132,133,134,135,136,137,138]. In this study, compared with the live bait fish diet, the pellet feed diet changed the intestinal colony structure of mandarin fish, which may be mainly caused by changes in the dietary structure and also be affected by environmental stress (including dietary stress) [139]. Disturbance of gut microbiota balance could lead to the establishment of harmful bacteria, causing disease problems [140,141,142,143]. In addition to diet, probiotic treatments can also affect the gut microbiome [144,145]. Probiotics play an essential role in the welfare of the host by maintaining a healthier balance of intestinal microbiota, which provides a defensive barrier against colonization of harmful bacteria and stimulates the immune system [146,147,148]. In this study, the addition of three probiotics significantly reduced the increased abundance of pathogenic bacteria *Aeromonas* caused by pellet-feed feeding, which may be achieved through the direct competition of probiotics on the abundance of pathogenic bacteria and indirect regulation of host immunity. According to reports, lactic acid bacteria inhibit the growth of harmful bacteria by producing antimicrobial compounds and competing for nutrients and attachment sites [41,149]. The present result agrees with earlier findings where a similar decrease in pathogenic bacteria (*Aeromonas* sp. and *Pseudomonas* sp.) was reported in giant freshwater prawn (*Macrobrachium rosenbergii*) feeding with a diet supplemented with *L. plantarum* [150]. This result is also consistent with the early discovery which reported that *L. rhamnosus* micro-granules administered for 30 days to tilapia larvae could significantly reduce the proportion of potentially pathogenic bacteria [38]. In addition, *C. butyricum* treatment reversed the increased abundance of intestinal pathogens in mice induced by severe acute pancreatitis and intra-abdominal hypertension [73]. All this indicates that these three probiotics can inhibit the abundance of harmful intestinal bacteria *Aeromonas* in the in vivo model resulting from direct competition between probiotics and pathogenic bacteria and host immunity regulation.

The intestine is the leading site of nutrient absorption, and the health of villi is a crucial factor influencing nutrient absorption. Consistent with the description of Wu et al. on the histological and histochemical characterization of mandarin fish tissues and organs, in our study, mandarin fish fed with live bait showed a conventional histological pattern of intestinal tissue [151]. In contrast, histological changes were detected in mandarin fish fed with pellet feed. Compared to mandarin fish fed on the live feed, the thickness of the foregut basement membrane in pellet feed-fed mandarin fish was significantly reduced, with similar results in other fish [88,152,153]. In addition, our results indicated that dietary supplement of *L. plantarum*, *L. rhamnosus* and *C. butyricum* enhanced the intestinal health development in mandarin fish by increasing the height of intestinal villi and the thickness of foregut basement membrane. Similarly, *L. plantarum* favorably recovered the cyclophosphamide-induced abnormal intestinal morphology in mice by improving the villus height [154]. Pangasius catfish (*Pangasius bocourti*) fed a diet supplemented with *L. plantarum* for 90 days exhibited a greater villus height in all intestines, with significant differences in the proximal intestine [155]. Wang et al. reported that *C. butyricum* increased the jejunal villus length and jejunal villus height to crypt depth ratio, while they decreased the jejunal crypt depth compared with those of the control and protected the intestinal villi morphology in a piglet model [114]. According to Sewaka et al., *L. rhamnosus* increased the villous height in the proximal, middle and distal parts of the intestine of juvenile red tilapia (*Oreochromis* spp.) [37]. Moreover, Casas et al. reported that the intestinal villus height of weanling pigs tended to increase as the dose of *C. butyricum* increased in the diet [94]. Our findings indicate that the application of probiotics could effectively promote the intestinal health of mandarin fish fed with pellet feed, which may benefit from repair of the intestinal microbial barrier. At the same time, the promoting effect of probiotics on intestinal health may be one of the reasons for the improvement of survival rate of mandarin fish fed with pellet feed.

## 5. Conclusions

In summary, the present results confirmed that the application of *L. plantarum*, *L. rhamnosus* and *C. butyricum* could significantly improve the eating disorders of mandarin fish caused by pellet-feed feeding, which expressed as significantly increased POPFR and survival rate. All of these may be related to the ability of probiotics to regulate gut microbiota, activate immunity, boost appetite, improve antioxidant capacity and protect intestinal tissues. This study explores the problem of eating disorders in non-mammals and tried to solve the eating disorders caused by pellet-feed feeding of mandarin fish by regulating gut microbiota using probiotics. In this study, the influence of probiotics intervention on eating disorders and its mechanism were studied using mandarin fish fed with pellet feed as a model. Due to the complex interactions between the gut microbiota, immune system, appetite and oxidative stress, the causal relationship between them needs to be further investigated. The conversion of pellet feed for mandarin fish has always been considered a global problem, and this study provides a new train of thought. More solutions, such as the application of other probiotics, prebiotics or immunostimulants, are worth investigating.

## Figures and Tables

**Figure 1 microorganisms-09-01288-f001:**
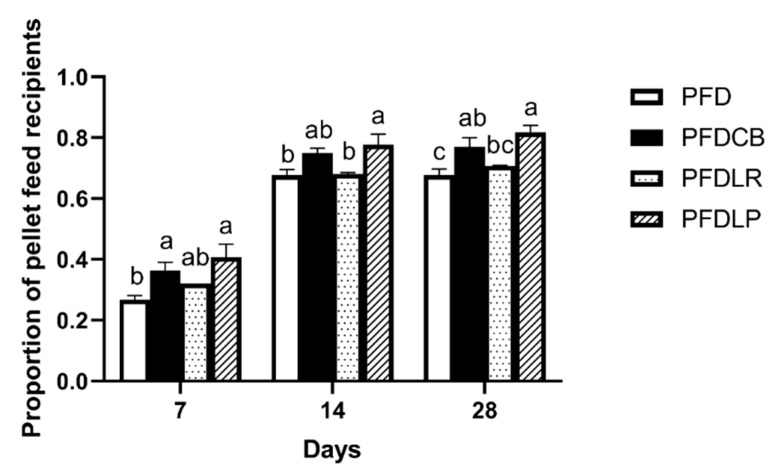
POPFR of mandarin fish in different feeding groups (PFD, PFDCB, PFDLR and PFDLP) at days 7, 14 and 28 of feeding. Data are presented as mean ± SEM (*n* = 3). Abbreviations: PFD, pellet-feed feeding group with probiotics free; PFDCB, pellet-feed feeding group with *C. butyricum* plus; PFDLR, pellet-feed feeding group with *L. rhamnosus* plus; PFDLP, pellet-feed feeding group with *L. plantarum* plus. A value followed by a lowercase superscript (a–c) differs significantly from all other values not followed by the same lowercase superscript at the same time point based on ANOVA followed by the post hoc test (*p* < 0.05).

**Figure 2 microorganisms-09-01288-f002:**
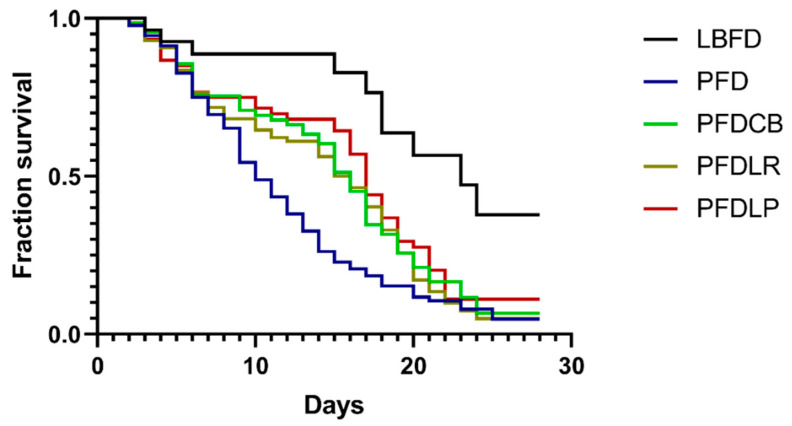
Kaplan Meyer’s (KM) survival analysis of mandarin fish in different feeding groups (LBFD, PFD, PFDCB, PFDLR and PFDLP) during 28 days of feeding. Abbreviations: LBFD, live bait fish feeding group; PFD, pellet-feed feeding group with probiotics free; PFDCB, pellet-feed feeding group with *C. butyricum* plus; PFDLR, pellet-feed feeding group with *L. rhamnosus* plus; PFDLP, pellet-feed feeding group with *L. plantarum* plus.

**Figure 3 microorganisms-09-01288-f003:**
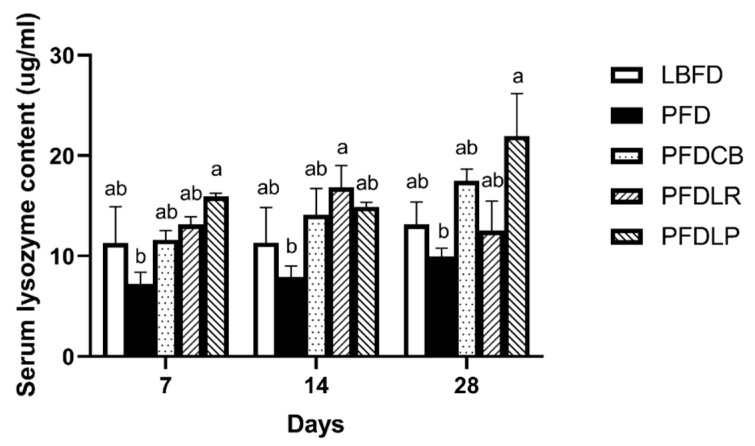
Serum lysozyme content of mandarin fish in different feeding groups (LBFD, PFD, PFDCB, PFDLR and PFDLP) at day 7, 14 and 28 of feeding. Data are presented as mean ± SEM (*n* = 9). Abbreviations: LBFD, live bait fish feeding group; PFD, pellet-feed feeding group with probiotics free; PFDCB, pellet-feed feeding group with *C. butyricum* plus; PFDLR, pellet-feed feeding group with *L. rhamnosus* plus; PFDLP, pellet-feed feeding group with *L. plantarum* plus. A value followed by a lowercase superscript (a–b) differs significantly from all other values not followed by the same lowercase superscript at the same time point based on ANOVA followed by the post hoc test (*p* < 0.05).

**Figure 4 microorganisms-09-01288-f004:**
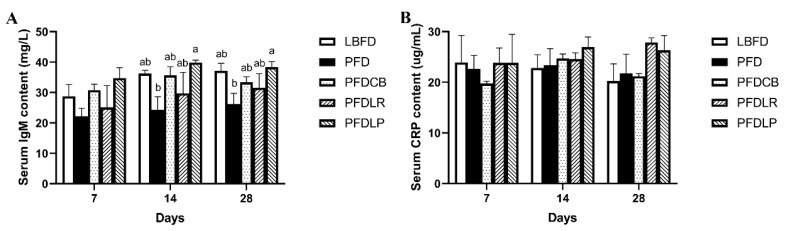
Serum content of IgM (**A**) and CRP (**B**) of mandarin fish in different feeding groups (LBFD, PFD, PFDCB, PFDLR and PFDLP) at day 7, 14 and 28 of feeding. Data are presented as mean ± SEM (*n* = 9). Abbreviations: IgM, immunoglobulin M; CRP, C-reactive protein; LBFD, live bait fish feeding group; PFD, pellet-feed feeding group with probiotics free; PFDCB, pellet-feed feeding group with *C. butyricum* plus; PFDLR, pellet-feed feeding group with *L. rhamnosus* plus; PFDLP, pellet-feed feeding group with *L. plantarum* plus. A value followed by a lowercase superscript (a–b) differs significantly from all other values not followed by the same lowercase superscript at the same time point based on ANOVA followed by the post hoc test (*p* < 0.05).

**Figure 5 microorganisms-09-01288-f005:**
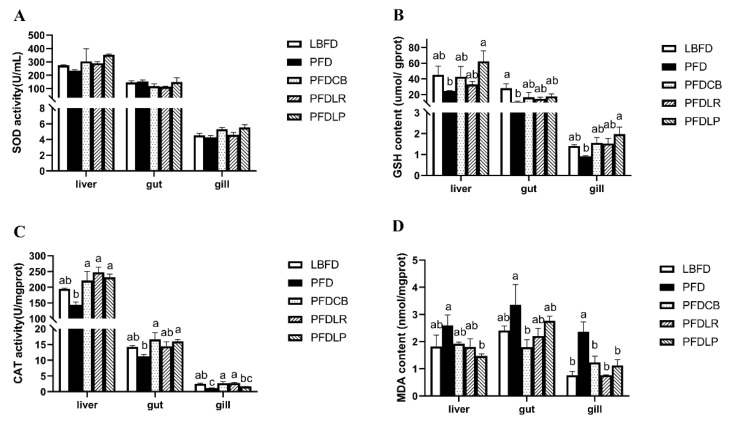
Activities of SOD (**A**), GSH (**B**) and CAT (**C**) and content of MDA (**D**) in the gut, liver and gills of mandarin fish in different feeding groups (LBFD, PFD, PFDCB, PFDLR and PFDLP) after being fed for 28 days. Data are presented as mean ± SEM (*n* = 9). Abbreviations: SOD, superoxide dismutase; GSH, glutathione; CAT, Catalase; MDA, malondialdehyde; LBFD, live bait fish feeding group; PFD, pellet-feed feeding group with probiotics free; PFDCB, pellet-feed feeding group with *C. butyricum* plus; PFDLR, pellet-feed feeding group with *L. rhamnosus* plus; PFDLP, pellet-feed feeding group with *L. plantarum* plus. A value followed by a lowercase superscript (a–c) differs significantly from all other values not followed by the same lowercase superscript at the same time point based on ANOVA followed by the post hoc test (*p* < 0.05).

**Figure 6 microorganisms-09-01288-f006:**
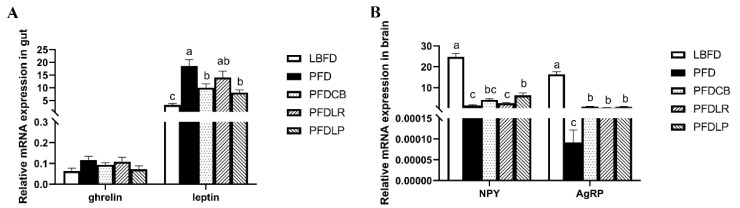
Relative mRNA expressions of appetite control genes in the gut (**A**) and brain (**B**) of mandarin fish in different feeding groups (LBFD, PFD, PFDCB, PFDLR and PFDLP) after being fed for 28 days. Data are presented as mean ± SEM (*n* = 9). Abbreviations: NPY, *nerve peptide y*; AgRP, *agouti gene-related protein*; LBFD, live bait fish feeding group; PFD, pellet-feed feeding group with probiotics free; PFDCB, pellet-feed feeding group with *C. butyricum* plus; PFDLR, pellet-feed feeding group with *L. rhamnosus* plus; PFDLP, pellet-feed feeding group with *L. plantarum* plus. Data are presented as mean ± SEM (*n* = 9). A value followed by a lowercase superscript (a–c) differs significantly from all other values not followed by the same lowercase superscript at the same time point based on ANOVA followed by the post hoc test (*p* < 0.05).

**Figure 7 microorganisms-09-01288-f007:**
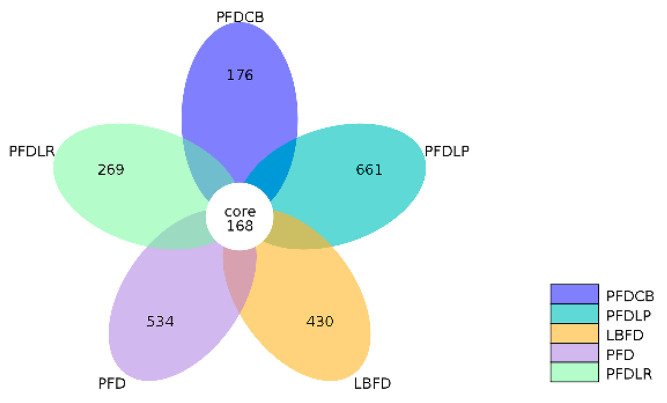
Venn diagram analysis depicting the numbers of shared and unique OTUs of mandarin fish intestinal microbial populations in different feeding groups (LBFD, PFD, PFDCB, PFDLR and PFDLP) after being fed for 28 days. Abbreviations: LBFD, live bait fish feeding group; PFD, pellet-feed feeding group with probiotics free; PFDCB, pellet-feed feeding group with *C. butyricum* plus; PFDLR, pellet-feed feeding group with *L. rhamnosus* plus; PFDLP, pellet-feed feeding group with *L. plantarum* plus.

**Figure 8 microorganisms-09-01288-f008:**
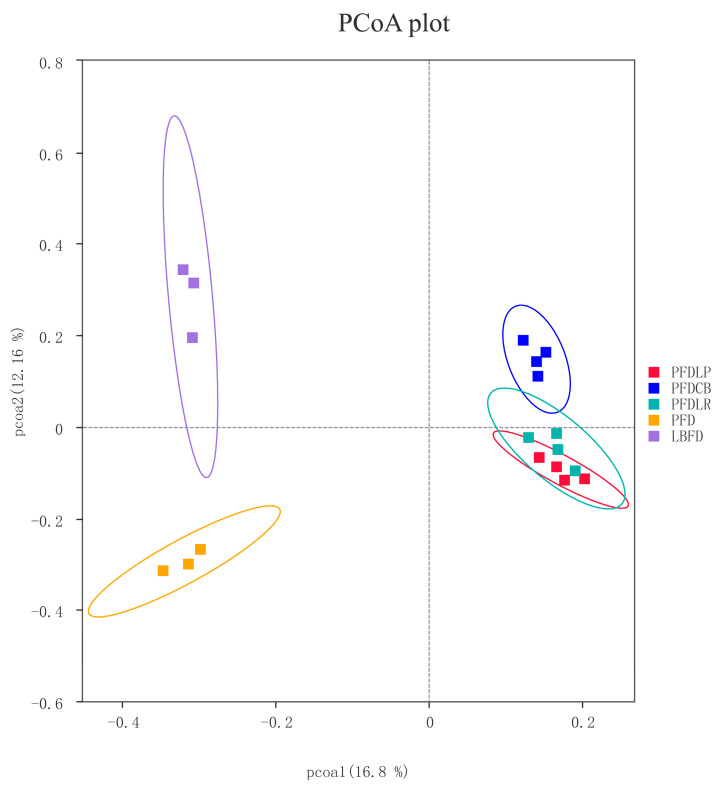
PCoA based on the binary jaccard distance of the intestinal bacterial communities of mandarin fish in different feeding groups (LBFD, PFD, PFDCB, PFDLR and PFDLP) after being fed for 28 days. Abbreviations: LBFD, live bait fish feeding group; PFD, pellet-feed feeding group with probiotics free; PFDCB, pellet-feed feeding group with *C. butyricum* plus; PFDLR, pellet-feed feeding group with *L. rhamnosus* plus; PFDLP, pellet-feed feeding group with *L. plantarum* plus.

**Figure 9 microorganisms-09-01288-f009:**
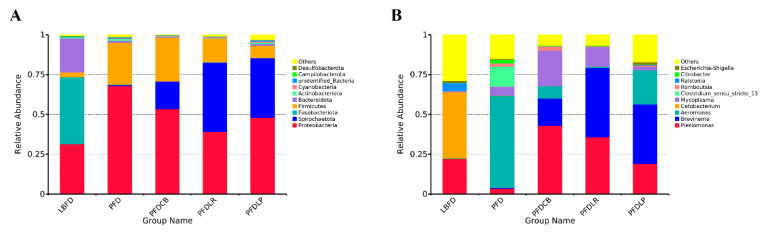
The abundance of composition at phylum (**A**) and genus (**B**) level in mandarin fish intestinal microbial populations in different feeding groups (LBFD, PFD, PFDCB, PFDLR and PFDLP) after being fed for 28 days. Abbreviations: LBFD, live bait fish feeding group; PFD, pellet-feed feeding group with probiotics free; PFDCB, pellet-feed feeding group with *C. butyricum* plus; PFDLR, pellet-feed feeding group with *L. rhamnosus* plus; PFDLP, pellet-feed feeding group with *L. plantarum* plus.

**Figure 10 microorganisms-09-01288-f010:**
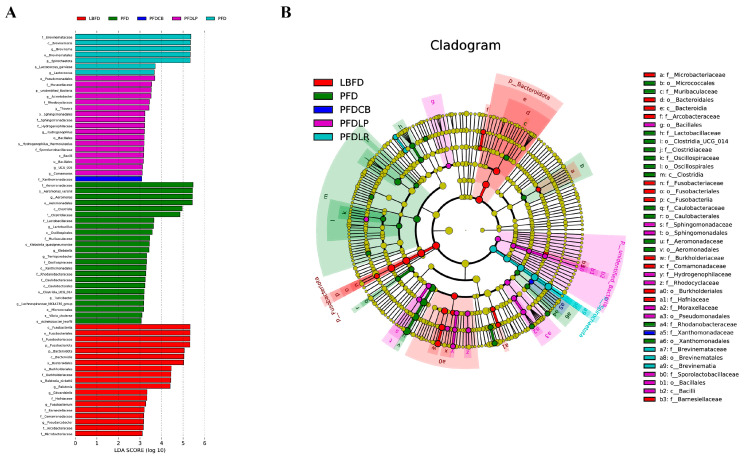
Intergroup variation in the relative abundance of the intestinal microbial communities. (**A**) Cladogram of LEfSe. (**B**) Bacterial taxa differentially displayed in the mandarin fish intestinal microbial populations in different feeding groups (LBFD, PFD, PFDCB, PFDLR and PFDLP) after being fed for 28 days were identified via LEfSe using an LDA score threshold of >3. Abbreviations: LBFD, live bait fish feeding group; PFD, pellet-feed feeding group with probiotics free; PFDCB, pellet-feed feeding group with *C. butyricum* plus; PFDLR, pellet-feed feeding group with *L. rhamnosus* plus; PFDLP, pellet-feed feeding group with *L. plantarum* plus.

**Figure 11 microorganisms-09-01288-f011:**
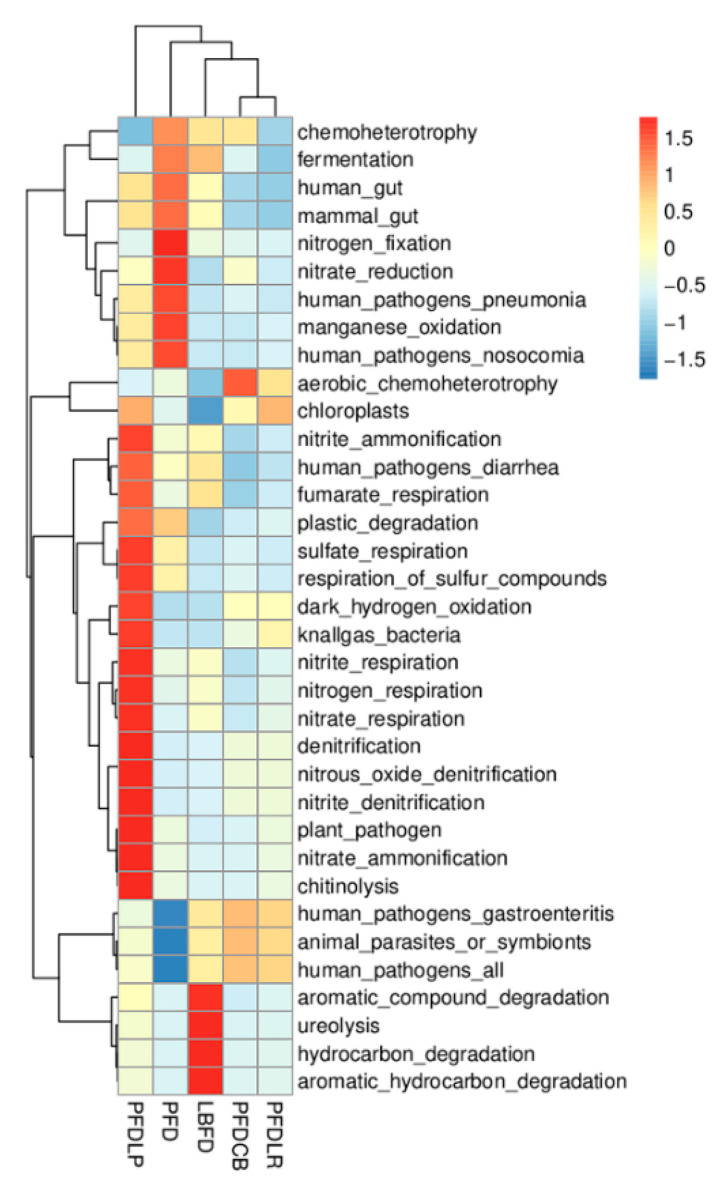
Heatmap showing the relative abundances of KEGG ortholog groups of mandarin fish intestinal microbial populations in different feeding groups (LBFD, PFD, PFDCB, PFDLR and PFDLP) after being fed for 28 days. The heatmap was made based on Tax4Fun functional annotations, and the color intensity indicates the abundance information. Abbreviations: LBFD, live bait fish feeding group; PFD, pellet-feed feeding group with probiotics free; PFDCB, pellet-feed feeding group with *C. butyricum* plus; PFDLR, pellet-feed feeding group with *L. rhamnosus* plus; PFDLP, pellet-feed feeding group with *L. plantarum* plus.

**Figure 12 microorganisms-09-01288-f012:**
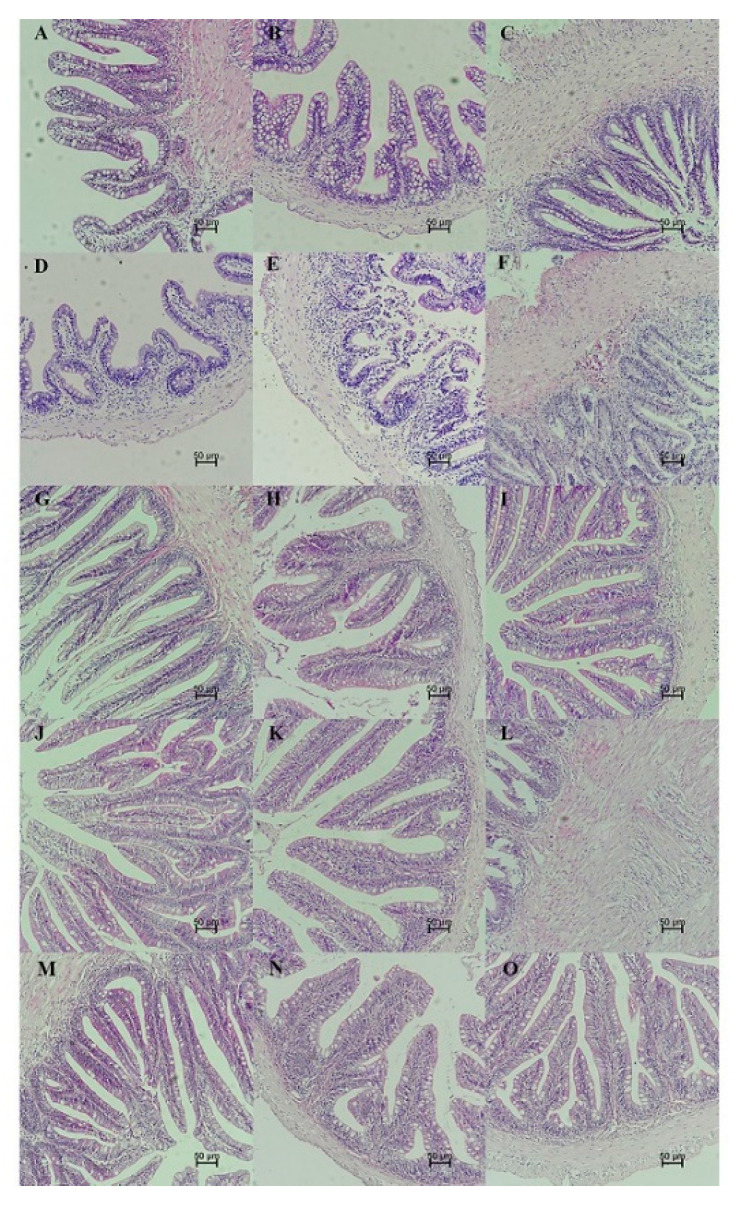
Photomicrographs showing histological sections of the intestinal tract of mandarin fish in different feeding groups (LBFD, PFD, PFDCB, PFDLR and PFDLP) after being fed for 28 days. (H.E. staining; scale bar: 50 μm; magnification ×200). (**A**–**C**) Foregut, midgut and hindgut of mandarin fish in LBFD group. (**D**–**F**) Foregut, midgut and hindgut of mandarin fish in PFD group. (**G**–**I**) Foregut, midgut and hindgut of mandarin fish in PFDCB group. (**J**–**L**) Foregut, midgut and hindgut of mandarin fish in PFDLR group. (**M**–**O**) Foregut, midgut and hindgut of mandarin fish in PFDLP group. Abbreviations: H.E., hematoxylin and eosin staining; LBFD, live bait fish feeding group; PFD, pellet-feed feeding group with probiotics free; PFDCB, pellet-feed feeding group with *C. butyricum* plus; PFDLR, pellet-feed feeding group with *L. rhamnosus* plus; PFDLP, pellet-feed feeding group with *L. plantarum* plus.

**Figure 13 microorganisms-09-01288-f013:**
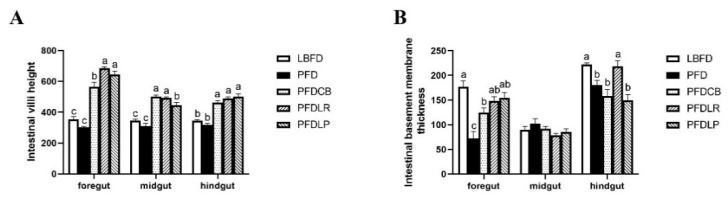
Intestinal villi height (**A**) and basement membrane thickness (**B**) of mandarin fish in different feeding groups (LBFD, PFD, PFDCB, PFDLR and PFDLP) after being fed for 28 days. Data are presented as mean ± SEM (*n* = 3). Abbreviations: LBFD, live bait fish feeding group; PFD, pellet-feed feeding group with probiotics free; PFDCB, pellet-feed feeding group with *C. butyricum* plus; PFDLR, pellet-feed feeding group with *L. rhamnosus* plus; PFDLP, pellet-feed feeding group with *L. plantarum* plus. A value followed by a lowercase superscript (a–c) differs significantly from all other values not followed by the same lowercase superscript at the same time point based on ANOVA followed by the post hoc test (*p* < 0.05).

**Table 1 microorganisms-09-01288-t001:** Sequences of primer pairs used for real-time quantitative PCR in this study.

Gene	Primer Name	Primer Sequence (5′-3′)	Annealing Temp (°C)
*ghrelin*	Scghrelin-F	GCTTTCTCAGCCCTTCAC	60
	Scghrelin-R	GGTTGTCCTCAGTGGGTTG	
*leptin*	scleptinB-F	CGAGAGTCACCTTTACCTG	58
	scleptinB-R	GTGCAAATAAGCCTCTAAGTG	
*npy*	scNPY-F	GCAAATCTCCCTCTGACAATC	60
	scNPY-R	GGTTTCACCGGGTATCCTT	
*agrp*	scAgRP-F	GAGCCAAGCGAAGACCAGA	58
	scAgRP-R	GCAGCACGGCAAATGAGAG	
*β* *-actin*	β-actin-F	CCCTCTGAACCCCAAAGCCA	59
	β-actin-R	CAGCCTGGATGGCAACGTACA	

**Table 2 microorganisms-09-01288-t002:** Richness and diversity indices of mandarin fish intestinal microbial populations in different feeding groups (LBFD, PFD, PFDCB, PFDLR and PFDLP) after being fed for 28 days.

Index	LBFD	PFD	PFDCB	PFDLR	PFDLP
Observed species	418.33 ± 107.84 ^c^	721.00 ± 66.55 ^a,b^	435.75 ± 18.91 ^c^	493.75 ± 85.17 ^b,c^	777.25 ± 35.03 ^a^
Shannon	2.61 ± 125.16 ^a^	2.85 ± 39.55 ^a^	1.88 ± 22.37 ^a^	1.62 ± 126.45 ^a^	2.85 ± 12.40 ^a^
Simpson	0.71 ± 151.35 ^a^	0.59 ± 47.19 ^a^	0.49 ± 16.06 ^a^	0.41 ± 131.67 ^a^	0.55 ± 29.14 ^a^
Chao 1	579.12 ± 0.51 ^b,c^	859.22 ± 0.10 ^a,b^	556.92 ± 0.34 ^c^	692.79 ± 0.35 ^a,b,c^	918.94 ± 0.33 ^a^
ACE	612.28 ± 0.07 ^b^	902.63 ± 0.07 ^a,b^	588.27 ± 0.12 ^b^	722.63 ± 0.12 ^a,b^	993.75 ± 0.05 ^a^
PD whole tree	68.41 ± 15.70 ^b^	85.20 ± 5.46 ^b^	80.21 ± 16.99 ^b^	169.72 ± 44.11 ^a,b^	199.00 ± 36.99 ^a^

ACE: abundance-based coverage estimator; LBFD, live bait fish feeding group; PFD, pellet-feed feeding group with probiotics free; PFDCB, pellet-feed feeding group with *C. butyricum* plus; PFDLR, pellet-feed feeding group with *L. rhamnosus* plus; PFDLP, pellet-feed feeding group with *L. plantarum* plus. The numbers represent the mean ± SEM (*n* = 3). A value followed by a lowercase superscript (a–c) differs significantly from all other values not followed by the same lowercase superscript at the same time point based on ANOVA followed by the post hoc test (*p* < 0.05).

## Data Availability

The data presented in this study are available on request from the corresponding author.

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
