# Peer review of "Probiotics Improve Eating Disorders in Mandarin Fish (Siniperca chuatsi) Induced by a Pellet Feed Diet via Stimulating Immunity and Regulating Gut Microbiota"

_microorganisms, 2021, doi:10.3390/microorganisms9061288_

Round 1

Reviewer 1 Report

Very interesting work and model for testing the potential probiotic effect of L. plantarum, L. rhamnosus and C. butyricum.
Check the new lactobacilli terminology and change accordingly. And harmonize the use of the term microbiota instead of microflora. Graphs and images are too small and no detail can be seen on them, so this should be corrected.

Reviewer 2 Report

The manuscript is about using probiotic in diet for mandarin fish to improve eating disorders and fish immunity. This manuscript is reasonable and suggested to be published in your esteemed journal after a minor revision as follows:

  1. The authors have to introduce the genes that being analyzed in this study in the introduction section.
  2. What kind of live bait used in this study?
  3. Why the concentration of 108 cfu/ml was used in this study?
  4. Did the authors determine the viability of probiotic in water after probiotic being added into water? How long will the probiotic survival in water?
  5. Fish were sampled for the physiological analysis during the experiment which might influence the results of growth performance. It is therefore suggested that the data of growth performance should be deleted.
  6. If the growth performance data being included in this study, feed efficiency should be presented.
  7. The results about the figures4-7, and table 3 should be re-addressed, such as the authors mentioned that GSH content and CAT activity were remarkable elevated in the liver, gut and gill of fish in probiotic supplemented groups as compared to the PFD group, but the CAT in gut of PFDLR, CTA in gill of PFDLP, GSH in liver of PFDLR, GSH in gut of all probiotic groups, and GSH in gill of PFDCB and PFDLR were not significantly different to the PDF. Same problem was also found in other data description.
